# Necrotizing Enterocolitis: A Comprehensive Review on Toll-like Receptor 4-Mediated Pathophysiology, Clinical, and Therapeutic Insights

**DOI:** 10.3390/biomedicines13092288

**Published:** 2025-09-17

**Authors:** Asuka Ishiyama, Hee-Seong Jang, Jay M. Dintaman, Johannes W. Duess, Cody Tragesser, Chhinder P. Sodhi

**Affiliations:** 1Department of Pediatric General and Urogenital Surgery, Johns Hopkins University School of Medicine, Tokyo 113-8421, Japan; askato@juntendo.ac.jp; 2Division of Pediatric Surgery, Department of Surgery, Johns Hopkins University School of Medicine, Baltimore, MD 21205, USA; johannes.duess@medizin.uni-leipzig.de (J.W.D.); ctrages1@jh.edu (C.T.); 3Department of Pediatrics, Uniformed Services University of the Health Sciences, Bethesda, MD 20814, USA; jay.dintaman@usuhs.edu; 4Department of Pediatric Surgery, University of Leipzig, 04103 Leipzig, Germany

**Keywords:** necrotizing enterocolitis, preterm infants, toll-like receptor 4, enteric nervous system, microbiome, dysbiosis, probiotics, formula feeding, intestinal barrier, inflammation, immune cells

## Abstract

This review integrates clinical, immunological, microbial, pathophysiological, and therapeutic perspectives on necrotizing enterocolitis (NEC)—a leading cause of morbidity and mortality in premature infants. We summarize the clinical burden and risk factors; elucidate key immune and cellular mechanisms, including TLR4 signaling, epithelial barrier dysfunction, and enteric nervous system involvement; and provide a concise overview of experimental models. We also highlight microbial dysbiosis, ischemia, multiorgan injury, and recent advances in pathogenesis, as well as current and emerging therapies such as probiotics, breast milk components, TLR4 inhibitors, and immunomodulators, emphasizing the need for a multidisciplinary approach to accelerate discovery and improve outcomes. Overall, this review bridges mechanistic insights to clinical applications and supports the pursuit of personalized NEC prevention.

## 1. Introduction

**Background & Current Understanding:** Necrotizing enterocolitis (NEC) remains one of the most devastating gastrointestinal emergencies in neonatology, presenting high morbidity, mortality, and healthcare burden. It primarily affects preterm infants, particularly those born before 32 weeks of gestation or weighing less than 1500 g. The incidence of NEC ranges from 2% to 7% in very low birth weight (VLBW) infants and accounts for as much as 15% of all neonatal intensive care unit (NICU) deaths [1]. In the United States alone, approximately 3000 new cases of NEC are reported each year. Although survival rates among preterm infants have improved, the incidence of NEC has stayed relatively stable over the past two decades, indicating that the increased survival has not yet led to a reduction in disease burden [1]. NEC disproportionately affects infants from socioeconomically disadvantaged backgrounds. Data reveal significantly higher incidence and severity among Black and Hispanic infants compared to White infants, infants in resource-limited NICUs, and formula-fed infants, especially in hospitals with lower access to donor human milk [2]. NEC remains one of the most expensive conditions in neonatal care, with recent estimates suggesting that the average cost per case of surgical NEC can exceed $300,000 to $500,000 [3,4]. Additionally, annual U.S. healthcare costs attributable to NEC are estimated to be between $500 million and $1 billion. Surgical NEC, in particular, results in prolonged hospital stays, requiring multiple surgeries, total parenteral nutrition (TPN) dependence, and intensive follow-up care. These figures exclude long-term disability care and indirect family costs. Survivors of NEC face significant long-term complications, including short bowel syndrome, feeding intolerance, growth failure, and neurodevelopmental delay affecting up to 40–60% of NEC survivors. There is also an increased risk of bronchopulmonary dysplasia (BPD) and cerebral palsy [5]. Emerging evidence links NEC to secondary organ injury in the brain and lungs, driven by systemic inflammation [6,7,8]. Globally, NEC is underdiagnosed and frequently fatal in low- and middle-income countries due to limited neonatal intensive care capacity, lack of access to sterile formula or human milk, and delayed diagnosis [9].

**Research Gap or Problem:** Despite decades of research, NEC remains a disease lacking reliable early diagnostic tools, universally effective preventive measures, or targeted curative therapies. Current management is mostly supportive, and most mechanistic insights (such as immune activation, microbial dysbiosis, epithelial barrier dysfunction) have been studied separately rather than within a combined translational framework. This fragmentation hinders progress in translating basic discoveries into clinical applications interventions.

**Purpose & Aim of the Review:** This review aims to synthesize current understanding of NEC by integrating perspectives from clinical medicine, immunology, microbiology, pathophysiology, and therapeutic development. Our main goal is to emphasize the multidisciplinary approach as the key contribution of this work—asserting that coordinated collaboration across different specialties is crucial to accelerate progress in diagnosis, prevention, and treatment.

**Scope of the Review:** We begin with the clinical burden, risk factors, and treatment strategies. We then explore pathophysiological mechanisms, including compromised gut epithelium, immune overactivation through TLR4 signaling, microbial dysbiosis, and systemic consequences such as brain and lung injury. Experimental models are summarized as the foundation for these mechanistic insights. Finally, we discuss current and emerging therapies—ranging from breast milk components and probiotics to immunomodulators and regenerative approaches—and outline future directions for prevention, treatment, and long-term care.

**Roadmap & Novel Contribution:** What distinguishes this review is its explicit focus on multidisciplinary integration. We emphasize that understanding NEC requires collaboration among neonatology, radiology, surgery, immunology, microbiology, nutrition, pulmonology, neurology, developmental pediatrics, and computational biology. **Neonatology and neonatologists**—first responders—are crucial in early NEC detection and management. As primary caregivers for preterm and sick neonates, they are responsible for recognizing early, nonspecific signs such as abdominal distension, feeding intolerance, increased gastric residuals, or stooling changes. Early recognition enables prompt diagnosis and the development of effective treatment strategies, including bowel rest [10], antibiotics [11], and surgical consultation [3,12]. They also guide the transition from observation to urgent intervention and coordinate multidisciplinary care involving radiology, surgery, infectious disease, gastroenterology, and nutrition (feeding strategies). **Radiology**, the radiologic imaging is crucial for diagnosing, staging, and monitoring NEC. Abdominal radiographs remain the primary method for detecting essential signs, such as pneumatosis intestinalis, portal venous gas, fixed dilated loops, and pneumoperitoneum—the presence or absence of these findings drives clinical staging and surgical decision-making [12,13,14,15,16]. Close collaboration among radiologists, neonatologists, and pediatric surgeons enhances diagnostic accuracy, optimizes intervention timing, and facilitates multidisciplinary decision-making. **Pediatric Surgery** is the key to survival as NEC can rapidly progress to intestinal perforation and peritonitis, necessitating urgent intervention. Pediatric surgeons face difficult decisions about when to perform laparotomy, bowel resection, or peritoneal drainage in critically ill infants. Surgical teams must also manage post-operative complications like short bowel syndrome and coordinate ongoing care, including TPN dependence [12]. **The hematologic and immunologic systems** offer vital insights into NEC development. Hematologic indicators, such as thrombocytopenia, neutropenia, and abnormal coagulation profiles, serve as early signs of disease severity and progression [17]. Systemic inflammatory markers (e.g., C-reactive protein (CRP), pro-inflammatory cytokine, Interleukin-6 (IL-6), and lactic acidosis frequently indicate systemic involvement and intestinal ischemia [17]. From an immunological standpoint, NEC results from an overactive innate immune response in the immature intestine. Abnormal activation of TLR4 on epithelial and immune cells by lipopolysaccharide (LPS), a component of the Gram-negative bacterial cell wall, initiates inflammation, cell death, and barrier dysfunction [18,19,20]. Excessive TLR4 signaling leads to overactivation of macrophages and Th17 cells, suppression of protective regulatory T cells, and abnormal neutrophil responses, which further damage the mucosa and contribute to systemic inflammation [21]. Insights from hematology and immunology inform the development of immunomodulatory and biomarker-based strategies for early diagnosis, treatment, and prognosis. **Microbiology** analyzes the intestinal microbiota, central to NEC pathology. The identification of dysbiotic patterns, increased microbial sialidase activity, and the roles of specific taxa (e.g., *Enterobacteriaceae*, *Bifidobacteriaceae, Lactobacilli*) have increasingly been demonstrated as significant components of NEC disease [22]. Microbiome-targeted interventions include probiotics, symbiotics, and bacteriophage therapy [23,24]. **Nutrition strategies** are crucial in promoting human milk, utilizing donor milk and human milk-based fortifiers, lipids, Human Milk Oligosaccharides (HMOs), and minimizing unnecessary exposure to formula [25,26,27,28,29]. Dietitians and neonatal nutrition experts also guide early versus delayed feeding strategies and prolonged parenteral nutrition for survivors of surgical NEC [10]. **Pulmonologists, neurologists, and developmental pediatricians** are increasingly involved in the follow-up care of patients recovering from NEC, as NEC can cause systemic inflammation, leading to injury to the developing brain and lungs. They investigate the mechanisms of the gut–lung axis [30] and the gut–brain axis [7,8], assist with pulmonary and neurocognitive assessments, and guide early rehabilitation strategies to enhance long-term outcomes. Finally, the **Biomedical engineers and computational scientists** contribute to NEC-on-a-chip systems, integrate multi-omics, and develop predictive models for early diagnosis. Their tools can simulate host–microbiome–immune interactions and identify high-risk infants before clinical symptoms appear [31,32].

By explicitly framing NEC through this multidisciplinary lens, we highlight not only the complexity of the disease but also the opportunities for synergy across fields. This integrative approach offers the potential for earlier diagnosis, personalized risk stratification, mechanism-guided therapies, and improved long-term outcomes for NEC survivors.

### 1.1. Clinical Aspects of NEC

**Epidemiology.** Preterm birth remains the most consistent and significant risk factor for developing NEC. Although the global incidence of NEC has not markedly declined over the past two decades, a prospective cohort study by Han et al. reported a notable reduction in the annual incidence of NEC in the United States, from 9% in 2006 to 6% in 2017 [33]. Both medically managed and surgically treated NEC cases showed decreases during this period (from 5.3% to 3.0% and from 3.4% to 3.1%, respectively). Correspondingly, mortality rates also declined for both groups: surgical NEC mortality dropped from 36.6% to 31.6%, and medical NEC mortality declined from 20.7% to 16.8% [33]. Interestingly, the use of Primary Peritoneal Drainage (PPD)—a less invasive resuscitative procedure—nearly doubled in surgical NEC cases, rising from 23.2% in 2006 to 47.3% in 2017, while the rate of laparotomy declined from 49.7% to 47.3% over the same period [3,33]. This trend suggests that broader adoption of PPD for critically ill infants unable to tolerate laparotomy may have contributed to improved survival rates. Furthermore, increased use of early enteral feeding with human milk in VLBW infants may also have played a role in reducing mortality across both surgical and medical NEC cases [10,33,34]. Racial and ethnic disparities in NEC incidence and outcomes are receiving growing attention in the United States. A multivariable logistic regression analysis by Jammeh et al. found that the odds of developing NEC were higher among non-Hispanic Black (Adjusted Odds Ratio [AOR] 1.06, 95% CI: 1.01–1.11, *p* = 0.027) and Hispanic (AOR 1.13, 95% CI: 1.06–1.19, *p* < 0.001) infants compared to non-Hispanic White infants. Additionally, the odds of death following NEC were significantly higher for both non-Hispanic Black (AOR 1.35, 95% CI: 1.15–1.58, *p* < 0.001) and Hispanic (AOR 1.31, 95% CI: 1.09–1.56, *p* = 0.003) populations [35]. Although this analysis did not adjust for indirect socioeconomic variables such as insurance status or the level of NICU involvement, other findings suggest that even when Hispanic infants with extremely low birth weight initiate and maintain breastfeeding at higher rates than non-Hispanic White infants, they still exhibit elevated NEC incidence. This indicates that while human milk feeding offers protective benefits, it alone cannot overcome the health disparities underlying the differences in NEC risk. Postnatal feeding differences intersect with numerous pre-, peri-, and postnatal risk factors for NEC, which are known to affect ethnic and racial groups unequally. Large population studies show that intrauterine growth restriction (IUGR), small for gestational age (SGA), and preterm birth are more common among Black and Hispanic patients [36,37,38]. The same applies to pre-eclampsia [39,40], maternal vitamin D deficiency [41], and chorioamnionitis [42]. Epigenetic factors, closely related to environmental cues such as diet, toxin exposures, and psychosocial stressors, may also contribute to racial disparities in NEC risk [43,44]. Despite improvements in clinical management and neonatal care, the lack of comprehensive demographic data and mechanistic understanding has limited our ability to significantly reduce NEC incidence and mortality. There is a pressing need to develop standardized diagnostic criteria that can be applied globally, based on accurate population-level data, detailed socioeconomic status, NICU care levels, and treatment practices. Such efforts will enable more precise epidemiological tracking and offer critical insights into the risk factors and pathophysiological mechanisms underlying NEC. Table 1 provides a summary of the burden of NEC, incidence, disparities, and economic impact, for a quick overview.

**NEC and Gestational Age.** Gestational Age (GA) is one of the strongest predictors of NEC risk, with the lowest GA infants exhibiting the highest incidence. Globally, an estimated 14.9 million babies (range: 12.3–18.1 million) are born preterm (<37 weeks’ GA), accounting for approximately 11.1% of all live births. This includes 10.4% born very preterm (28 to <32 weeks) and 5.2% born extremely preterm (<28 weeks) [50]. NEC affects approximately 2% to 7% of infants born before 32 weeks’ GA [45], and nearly 90% of all NEC cases occur in infants born prior to 32 weeks. Interestingly, using Bell’s staging (Stage ≥ 2), the incidence of NEC is reported to be higher in infants born at 24 weeks compared to those born at 23 weeks GA [45,51]. Resuscitation practices of periviable infants < 24 weeks have changed over the previous two decades, resulting in increased numbers of infants born between 22 and 24 weeks receiving active post-natal care [52]. Survivability of this population in the U.S. has also increased from 18.4% in 2007 to 31.9% in 2018 with NEC prevalence increasing significantly from approximately 1% to 8% during the same period, suggesting that one contributor to the current NEC burden may be the increasing numbers of periviable infants receiving active care within high-resource NICUs. International comparisons show marked geographic variation in NEC incidence. Among infants born at <28 weeks GA, Japan reports the lowest NEC rate (2.0%; 95% CI: 1.8–2.2), whereas California, USA reports the highest (9.6%; 95% CI: 9.1–10.1). Similarly, for infants born at 28–31 weeks GA, NEC rates range from 0.2% (95% CI: 0.07–0.4) in Japan to 4.4% (95% CI: 4.0–4.7) in California [51]. Several factors appear to contribute to this regional disparity, perhaps most notably the higher usage of maternal and donor breast milk in Japanese NICUs. Mortality is significantly higher among infants with NEC compared to those without it (28% vs. 13%, *p* < 0.001) [35]. The second and third trimesters are critical for the anatomical and functional maturation of the gastrointestinal tract. In particular, infants born before 32 weeks often exhibit immature gastric electrical activity and delayed gastric emptying, impairing their ability to tolerate enteral feeds. Adequate gut motility for nutrient absorption typically does not mature until 29–34 weeks of gestation [49]. Altogether, this functional immaturity likely contributes to the heightened susceptibility of preterm infants to NEC.

**NEC and Birth Weight.** Birth weight is inversely linked to the incidence of NEC, as shown in large multicenter cohort studies [46,47], and is widely recognized as an independent risk factor for NEC development. Infants with extremely low birth weight (ELBW; <1000 g) and very low birth weight (VLBW; <1500 g) are at the highest risk [1]. The connection between birthweight and NEC risk remains even after controlling for gestational age, placental blood flow measures, perinatal factors, and clinical variables [48,53]. However, the association between birthweight and NEC is clearly influenced by geographic region and resource availability. Although the overall incidence of NEC among VLBW infants has been reported to be approximately 10%, this rate varies significantly across different countries and healthcare settings [1,54]. Using Bell’s stage ≥ 2 criteria, international data show that NEC incidence in VLBW infants is lowest in Japan (1.6%; 95% CI: 1.3–1.8) and highest in Poland (8.7%; 95% CI: 6.9–10.5). Among infants weighing less than 1000 g, the highest NEC rate has been reported in Finland (22%; 95% CI: 17.1–26.7) [45]. However, these data are limited to high-income countries and do not reflect the true global burden of NEC. To address this gap, Alsaied et al. conducted a systematic review and meta-analysis of cohort studies—including those from low- and middle-income countries—to estimate the global incidence of NEC in VLBW infants [1]. Using the Quality Effects Model to account for study heterogeneity, they estimated a global pooled incidence of 6.0% (95% CI: 4.0–9.0%). In the United States, extremely low birth weight (ELBW) infants (<1000 g) have a higher chance of receiving surgical procedures than VLBW infants, reflecting poor progress and vulnerability in ELBW infants [33]. Llanos et al. conducted population-based studies on the epidemiology of NEC from 1991 to 1998 in the United States and reported that infants weighing 750–999 g at birth had the highest incidence of NEC. The lower incidence among those born under 750 g presumably reflects a lower survival rate before developing NEC, which implies that improved survival rates in the future may lead to an increased incidence of NEC [55]. Overall, the timing of NEC diagnosis and surgical intervention is critical for rescuing VLBW infants, especially those born under 1000 g.

**Clinical Diagnosis.** The clinical diagnosis of NEC remains challenging due to the rapid progression of symptoms from subtle, non-specific early signs—such as feeding intolerance and worsening apnea of prematurity—to more overt and advanced manifestations, including bilious emesis, bloody stools, pneumatosis intestinalis, and bowel perforation. Among radiographic findings, the pathognomonic finding is pneumatosis intestinalis, reflecting intramural gas within the bowel walls and typically observed on plain abdominal radiographs. Serial radiographs over several hours can improve the detection of subtle pneumatosis in the early stages of disease progression [56]. Bedside ultrasonography is gaining recognition as a valuable, non-invasive, and cost-effective diagnostic modality. It enables real-time assessment of bowel wall integrity, thickness, peristalsis, and perfusion—especially when combined with color Doppler imaging—along with detection of pneumatosis [57,58]. However, the lack of NEC-specific laboratory biomarkers further complicates early diagnosis. The most commonly used laboratory markers, such as elevated C-reactive protein (CRP), leukocytosis, or metabolic acidosis, may be present but are non-specific and can also occur in other gastrointestinal conditions like meconium ileus or systemic inflammatory response states [17]. The most widely used system of clinical disease staging is Bell’s criteria, which consider clinical, radiographic and gastrointestinal findings [59]. Early disease is designated Stage I (Suspected NEC), correlating to non-specific clinical signs, normal or mild ileus on radiographs, and mild abdominal distension. Stage II (Definitive NEC) is defined by focal or widespread pneumatosis intestinalis and systemic signs of inflammation; Stage III (Advanced NEC) is diagnosed by peritonitis on examination and pneumoperitoneum indicating perforation. The utility of Bell’s criteria, however, has been criticized due to variability in disease presentation and its inability to clearly distinguish NEC from spontaneous intestinal perforation (SIP), which shows the same clinical symptoms as NEC with a single perforation in the terminal ileum [60]. Several alternative diagnostic frameworks have been proposed, including the Vermont Oxford Network (VON) criteria [61], the Centers for Disease Control (CDC) and Prevention definition [62,63], and the Gestational Age-Specific Case Definition of NEC (UKNC-NEC) [64]. The VON definition incorporates surgical and post-mortem findings alongside clinical and radiographic criteria derived from Bell’s staging, and it excludes cases of SIP—an exclusion that has been credited with contributing to the observed decline in NEC incidence in the United States. The CDC criteria are similar to the VON definition but do not explicitly exclude SIP. The UKNC-NEC criteria are distinctive in providing a gestational age–specific case definition, developed through population-based studies. In comparative evaluations, UKNC-NEC demonstrated a sensitivity of 63.9% (95% CI: 60.6–67.0) and a specificity of 96.8% (95% CI: 96.1–97.4), versus the VON definition’s sensitivity of 49.7% (95% CI: 46.3–53.0) and specificity of 97.2% (95% CI: 97.6–97.8) when applied to the same population cohort [64]. Emerging diagnostic modalities such as near-infrared spectroscopy (NIRS) are also being explored. Heide et al. reported that a high splanchnic-to-cerebral oxygenation ratio—calculated using NIRS—may be an early indicator of NEC in preterm infants. However, larger, multicenter studies remain to be validated for their clinical utility and reproducibility [65].

**NEC Biomarkers and Early Detection.** Efforts are being made to identify biomarkers that predict NEC development, including microbial signatures (via 16S or metagenomic sequencing), cytokines (e.g., IL-6, IL-8, LCN2), metabolites (e.g., bile acids, Short Chain Fatty Acids), and DNA methylation patterns. Point-of-care platforms that combine these markers with clinical data could enable real-time NEC risk prediction and early intervention [17,66].

### 1.2. Clinical Treatment Strategies

**Supportive and Medical Management.** Initial management of suspected NEC focuses on supportive care. This includes making the infant Nil Per Os (NPO) or reducing enteral feeds, initiating parenteral nutrition, placing a nasogastric tube for bowel decompression, and administering intravenous fluids. Broad-spectrum antibiotics are commonly started at the early stage of suspicion. However, the choice of antibiotic combinations, routes of administration, and treatment duration often varies by clinician, as there is insufficient evidence to establish standardized guidelines [67]. Due to the rapid and often devastating progression of NEC, additional supportive therapies are essential and may include mechanical ventilation to prevent hypoxia caused by restrictive lung disease or diaphragmatic elevation from abdominal distension, and hemodynamic support with inotropes and vasopressors to prevent cardiovascular collapse in severe cases of systemic inflammation or septic shock.

**Surgical Management.** Surgical intervention remains the definitive therapy for critically ill infants with NEC, particularly in the presence of intestinal perforation, which is considered the only absolute indication for surgery. Early perforation signs can be subtle, but clinical deterioration may occur quickly due to inflammation. Therefore, early involvement of a pediatric surgeon with clear clinical criteria for surgical intervention is essential. Surgery timing and approach are variable, and the surgeon’s judgment and experience, infant weight, and hemodynamic status must be considered, among other factors. PPD is a less invasive bedside procedure that entails the insertion of a drainage catheter through a small abdominal incision. Hemodynamically unstable or VLBW infants often undergo primary peritoneal drainage (PPD), though this is rarely the final intervention, and primary laparotomy should be considered for definitive control if it can be tolerated [68,69,70]. Exploratory laparotomy enables direct visualization of the bowel, allowing for resection of all necrotic segments, washout of the peritoneal cavity, creation of enterostomies, and closure if there are no marginally viable segments of intestine. If there is any doubt about the viability of the unresected intestine, a planned second-look operation should be considered within 24–48 h. Intestine with reversible injury may improve at second look, allowing the surgeon to preserve viable bowel and prevent short bowel syndrome. Among infants undergoing bowel resection for NEC, approximately 80% receive an enterostomy, while the remaining 20% undergo primary anastomosis [71]. Despite the higher frequency of enterostomy, large retrospective studies have shown that primary anastomosis yields comparable outcomes in terms of mortality, length of hospital stay, and time to full enteral feeds [71]. However, a prospective, randomized clinical trial is still needed to validate differences between these two surgical approaches.

### 1.3. Feeding Practices

**Historical perspective.** Despite significant advances in neonatal care, NEC still causes high morbidity and mortality in preterm infants. Prevention strategies and translational breakthroughs in NEC treatments are urgently needed and increasingly guided by mechanistic insights from both clinical and experimental research [72]. Historically, the immature gastrointestinal tract in preterm or VLBW infants was thought to be unable to digest and absorb nutrients effectively, contributing to the belief that early enteral feeding might cause feeding intolerance or functional ileus. As a result, TPN was commonly used to support these infants until gut maturation occurred [10,73]. However, more recent evidence indicates that early initiation and advancement of enteral feeds do not significantly increase the risk of NEC occurrence [34].

**Early versus delayed enteral feedings.** The timing of enteral feeding initiation in preterm infants has long been a focus in NEC prevention strategies. Historically, concerns about feeding intolerance and functional ileus in the immature gut caused the delayed start of enteral feeds, relying instead on total parenteral nutrition (TPN) until bowel maturity was achieved. In a multicenter retrospective cohort of 295 ICU patients, early enteral feeding was associated with significantly lower 28-day mortality (25.5% vs. 50.0%), decreased need for mechanical ventilation (66.5% vs. 80.6%), and shorter ICU stays (13.1 ± 16.4 vs. 16.2 ± 13.6 days). These benefits were especially evident in surgical patients, where delaying feeding was independently linked to higher mortality risk and longer ICU stays after adjusting for confounders. These findings support guidelines recommending early enteral nutrition for hemodynamically stable infants, using a gradual advancement approach to reduce complications [74].

**Human Milk.** Breast milk remains the most effective known intervention for reducing NEC incidence. It provides immunomodulatory and anti-inflammatory components (e.g., lactoferrin, sIgA, cytokines), growth factors (e.g., EGF, TGF-β) that promote mucosal healing [25,28,29,75,76], and human milk oligosaccharides (HMOs) that influence microbial colonization and suppress TLR4 signaling [26,77]. The timing of Mother’s Own Milk (MOM) initiation also influences NEC risk. Initiating MOM within the first seven postnatal days is associated with a significantly lower risk of NEC than later initiation. The absolute risk difference was −0.88% (95% CI: −1.15 to −0.61), and the relative risk of NEC was 0.69 (95% CI: 0.60–0.78). Donor milk (DM), although inferior to maternal milk in some components, still offers protective benefits compared to formula [29]. Fortification strategies are being optimized to provide adequate caloric intake while supporting gut health, especially in extremely preterm infants. These components are typically more concentrated in MOM than in donor human milk (DHM), which is often sourced from mothers of full-term infants [76,77].

**Fortified Human Milk.** Because MOM alone may not meet the high nutritional demands of preterm infants, it is often fortified with essential nutrients (Fortified Human Milk, FHM). In a randomized study comparing FHM with exclusive formula feeding, the incidence of NEC was significantly lower in the FHM group (1.6% vs. 13%, *p* ≤ 0.01) [78]. Human milk fortifiers that are bovine or human-based are both available, but bovine-based fortifiers remain the predominant current choice [79]. A recent Cochrane review by Premkumar et al. found insufficient evidence to support human milk-based fortification over bovine-based options and highlighted the need for a high-quality study comparing these two options in exclusively breast milk-fed neonatal populations [80]. Donor Human Milk (DHM) may serve as a reasonable substitute when MOM is insufficient, although consensus on its efficacy in preventing NEC is still evolving [28,78].

**Bovine Colostrum.** Bovine colostrum (BC), a nutrient-rich secretion from cows during the first days post-parturition [81], has been recently investigated as a supplement for the early feeding of very preterm infants (<32 weeks’ gestation), particularly when maternal milk (MM) is insufficient [82]. The PreColos randomized controlled trial evaluated whether BC supplementation could improve feeding outcomes compared to preterm formula (PF). Although BC contains high levels of protein and bioactive factors that promote gut maturation in preclinical models, the study found no significant reduction in time to full enteral feeding (120 mL/kg/d, TFF120) with BC use. Clinical safety was confirmed, but outcomes varied across neonatal units, suggesting that BC’s efficacy may depend on feeding practices and the proportion of MM available [82].

**Infant formula and NEC Risk.** Formula feeding, especially with bovine-based preparations, is consistently linked to a higher incidence of NEC compared to human milk. Exclusive formula feeding can double the risk of NEC in preterm infants, partly because of its effect on gut microbiota composition. Formula-fed infants show greater microbial diversity but have fewer beneficial bacteria like Bifidobacterium and Bacteroides, along with increased levels of Clostridium difficile. These changes lead to a pro-inflammatory intestinal environment, weakened mucosal immunity, and higher vulnerability to bacterial translocation. Animal studies also indicate that bovine-based formula speeds up antigen-presenting cell recruitment, decreases regulatory T-cell populations, and boosts B-cell activity. Although modern formulas are now supplemented with bioactive compounds such as lactoferrin, human milk oligosaccharides, and prebiotics to more closely imitate human milk, these modifications have not yet eliminated the disparity in NEC risk [83].

## 2. Pathogenesis and Mechanistic Insights

NEC arises from a cascade of aberrant pathophysiological events culminating in sudden and severe intestinal inflammation and necrosis, predominantly affecting the premature intestine. The initial NEC trigger is believed to involve the invasion of pathogenic bacteria into the immature intestinal barrier, resulting in a dysregulated pro-inflammatory immune response. This pro-inflammatory immune activation triggers the uncontrolled release of cytokines, such as IL-6, IL-1β, TNF, and reactive oxygen species (ROS), leading to widespread epithelial damage through apoptosis, necroptosis, and necrosis [18,84,85]. Clinically, this process manifests as acute intestinal inflammation and can progress to perforation, peritonitis, systemic sepsis, and mortality [85]. Several intrinsic vulnerabilities of the premature host predispose it to this destructive cascade. These include immaturity of the intestinal barrier and immune system, microbial dysbiosis, exaggerated TLR4 signaling, impaired microvascular perfusion, and altered cellular repair mechanisms. Increasing evidence using mouse studies and human biospecimens indicates that NEC-induced intestinal inflammation can extend beyond the gut, with systemic inflammatory signals contributing to secondary organ injury, particularly in the lungs and brain [6,8]. These remote effects are of primary clinical concern, given their contribution to the long-term morbidity observed in NEC survivors. A wide array of experimental models—involving small animals (mice, rats, Quails), large animals (Sheep, Non-human primates, and preterm piglets), and emerging ex vivo and in vitro systems—have been developed to simulate the multifactorial nature of NEC. These models incorporate key features such as prematurity, enteric bacterial colonization, exposure to lipopolysaccharide (LPS), formula feeding, and hypoxic or cold stress [19,84,86]. They continue to serve as essential platforms for investigating disease mechanisms and testing candidate therapies on a large scale.

### 2.1. Experimental Models to Study the Pathogenesis of NEC 

The following section outlines experimental NEC models that have been instrumental in elucidating TLR4-driven mechanisms described above. Because of the ethical and logistical limits of studying early NEC development in human infants, experimental models have been essential in uncovering the molecular, microbial, immunological, and vascular factors involved in the disease pathogenesis. These models generally include in vivo small and large animal models, as well as in vitro and ex vivo systems, each providing distinct advantages and disadvantages. Recent reviews by Kovler et al. [15] and Bautista et al. [87] offer an in-depth analysis of the experimental NEC models. While no model perfectly recapitulates the complexity of human NEC, several reproduce key pathological and immunological features—particularly those relevant to TLR4-mediated injury. Below, we summarize the most widely used models, followed by a comparative table aligning their histopathological and clinical features with human NEC. The experimental NEC models described below and in Table 2 represent current approaches to studying the pathophysiology of NEC.

**Mouse Models of NEC:** Experimental mouse models of NEC are critical for understanding the molecular and cellular mechanisms underlying NEC pathophysiology. They are widely used due to their genetic manipulation and the availability of knockout strains from multiple resources at a relatively low cost. These models enable genetic manipulation and mechanistic studies of key pathways such as TLR4 signaling, epithelial cell death, immune activation, and barrier dysfunction. The most common model involves formula feeding, intermittent hypoxia, cold stress, colonization with opportunistic bacteria, or exposure to LPS in neonatal mice, replicating important disease features, primarily in the terminal ileum, similar to the human condition [15,18]. Importantly, research has demonstrated that NEC pathology in mice depends on TLR4 signaling, and modifications to the model—including bacterial supplementation, cytokine sensitization, or use of humanized flora—have further improved its translational relevance. Although mouse models cannot fully capture the complexity of human NEC, they offer a powerful tool for high-throughput screening of potential therapies and validating mechanistic hypotheses in vivo.

**Rat Models:** Rat models have traditionally been essential for studying NEC, especially in understanding the roles of feeding, hypoxia, and microbial colonization in disease development [15,88]. The classic NEC model in neonatal or newborn rats involves gavage feeding with formula, combined with exposure to intermittent hypoxia and cold stress over several days [15,22]. This model consistently causes intestinal injury marked by mucosal necrosis, inflammation, and pneumatosis, notably in the distal small intestine. Benefits of using rats include ease of handling, well-understood developmental physiology, and enough tissue for biochemical and histological studies. Although genetic manipulation is more limited in rats than in mice, rat NEC models have played a crucial role in identifying key factors in disease development, such as oxidative stress, nitric oxide imbalance, bacterial translocation, and inflammatory cytokine activation. These models remain important for testing nutritional and drug-based treatments in a preclinical setting.

**Preterm Piglet Model:** The preterm piglet model is widely regarded as the most physiologically and clinically relevant large animal model for NEC [15,87,89]. Piglets are delivered via cesarean section at approximately 90–95% gestation, then resuscitated, catheterized, and formula-fed under controlled conditions to induce NEC-like pathology [77,90]. This model closely mirrors the human preterm infant in terms of gastrointestinal anatomy, intestinal maturation, digestive enzyme expression, and microbiota colonization. Importantly, it enables the evaluation of clinical parameters such as abdominal distension, gastric residuals, bloody stools, apnea, and systemic sepsis—features difficult to reproduce in rodent models. The piglet model offers unique advantages for translational research, including the ability to monitor vital signs, collect serial blood and tissue samples, and test interventions requiring substantial dosing or repeated administration. It has been pivotal in advancing our understanding of the effects of feeding strategies (e.g., colostrum vs. formula), red blood cell transfusions, iron supplementation, and microbial interventions, including probiotics and human milk oligosaccharides. Furthermore, piglets allow for endoscopic and imaging techniques that are not feasible in smaller animals. Despite its strengths, the piglet NEC model is resource-intensive, requiring surgical expertise, intensive care infrastructure, workforce, and extremely laborious procedures, as well as high animal care costs. Moreover, its use is constrained by the limited availability of genetic tools and a lack of transgenic pig strains for mechanistic interrogation. Nonetheless, the preterm piglet remains indispensable for bridging preclinical findings to human trials, particularly in the context of nutritional and therapeutic interventions for NEC.

**Quails NEC Model:** The gnotobiotic quail model provides a unique, spontaneous way to study NEC that reflects several key features of human disease without requiring hypoxic or hypothermic stress [91]. This model uses germ-free neonatal quails that are orally colonized with *Clostridium butyricum*, a commensal bacterium isolated from a fatal human NEC case, and they are fed a lactose-containing diet. Due to their natural alactasia and anatomically prone ceca to stasis, quails are especially susceptible to undigested lactose fermentation. The model mimics essential NEC characteristics such as cecal wall thickening, pneumatosis, hemorrhage, and mucosal necrosis. This model is especially valuable for studying the roles of bacterial metabolites, host–microbiota interactions, and innate immune activation in NEC. However, limitations include species-specific immune features and anatomical differences that may limit translational applicability beyond microbiota-driven mechanisms.

**Pre-Term Rabbit Model of NEC:** The preterm rabbit model provides a novel, physiologically relevant platform for studying NEC in a non-rodent, non-invasive system that closely mimics NICU conditions. Preterm New Zealand white rabbits are delivered via cesarean section two days before term and exposed to a combination of enteral feeding, bacterial colonization (*Enterobacter cloacae*), and pharmacological agents (ranitidine, indomethacin) [92]. A key innovation in this model is the use of intermittent or complete anal blockage with tissue adhesive to simulate intestinal dysmotility and distension—factors associated with NEC pathogenesis. The severity and incidence of NEC-like injury correlate with the duration and completeness of the blockage, providing a flexible system to model progressive disease. Histopathological changes mirror human NEC, ranging from villous tip sloughing to transmural necrosis. This model supports the evaluation of therapeutic strategies such as probiotics and acidified formulas in a preterm gastrointestinal context. Advantages include controlled delivery of NEC risk factors and precise histological grading. However, limitations involve moderate survival rates under complete blockage conditions and limited genetic tools compared to rodent models. Nonetheless, the rabbit NEC model adds valuable mechanistic and translational insight, particularly for interventions targeting intestinal motility, barrier function, and microbial translocation.

**Non-human Primates Model of NEC (*Rhesus macaques*):** Non-human primate (NHP) models offer unparalleled physiological relevance for studying NEC, especially in the context of extreme prematurity, infection, and neurodevelopmental outcomes. In rhesus macaques, chronic intra-amniotic infection with *Ureaplasma parvum* induces preterm birth, systemic fetal inflammation, and gastrointestinal morbidity, including NEC [93]. One of the infant monkeys in the infection cohort that developed clinical NEC and did not respond to standard care, mirroring the clinical course of the human disease. NHP models, with over 95–98.5% genetic similarity to humans [94], are valuable for studying intrauterine inflammation, ventilation, parenteral nutrition, neurobehavioral development [104], and NEC in the context of prematurity-related conditions like bronchopulmonary dysplasia and white matter injury [93]. Although resource-intensive and ethically complex, NHP models represent the gold standard for translational NEC research, bridging rodent models and human clinical trials. Despite all clinically relevant advantages, the NHP NEC models are uncommon and almost non-existent.

**Non-human Primates Model of NEC (Baboon):** The preterm baboon model serves as a highly translational, non-human primate system for studying NEC in the context of extreme prematurity. Premature baboons delivered at 125 days’ gestation (67% of the term, equivalent to 26–27 weeks in humans) and managed with neonatal intensive care protocols develop spontaneous NEC with clinical, radiologic, and histopathologic features remarkably similar to those of human disease [95]. These features include feeding intolerance, abdominal distension, pneumatosis intestinalis, and mucosal necrosis. This model enables invasive monitoring, longitudinal sampling, and detailed mechanistic studies, thanks to its close genetic and physiological similarity (~96%) to humans. Despite its high cost and logistical challenges, the baboon NEC model offers an unmatched platform to link rodent research with clinical findings, especially in studying prematurity-driven immune and epithelial response dysfunction.

**Epithelial Cell Lines NEC Models:** Recent advances in NEC modeling have drawn increased attention to in vitro systems that mimic essential intestinal epithelial cells (IEC) and immune functions. While IEC lines such as Caco-2, HT-29, IEC-6, and H4 have been extensively utilized for decades [96,97,98], each possesses unique advantages and limitations. These cell lines can be obtained from the American Type Culture Collection (ATCC)—Caco2 (HTB-37™), HT-29 (HTB-38™), IEC6 (CRL-1592™), and H4 (HTB-148™)—and are easy to culture and study. Caco-2 and HT-29 enable the study of intestinal barrier proteins; however, they lack complete goblet cell differentiation and mucus production. Fetal human FHs 74 and H4, derived from human fetal intestine, provide more developmentally relevant insights [99,100]. These cell lines have been used to model NEC injury by exposing them to Lipopolysaccharides (LPS), Hydrogen peroxide (H_2_O_2_), pro-inflammatory cytokines (e.g., TNF-α, IFN-γ), or pathogens, such as human fecal bacteria from babies diagnosed with severe NEC [18,101].

**Intestinal Organoids NEC models:** Intestinal organoids with three-dimensional epithelial structures derived from Lgr5^+^ stem cells enable the recapitulation of the crypt-villus axis and stem cell activity. They have been generated from both human and mouse intestines and can be exposed to NEC-associated stressors such as LPS, hypoxia, or bacterial toxins, reproducing NEC-like features in a dish, referred to as “NEC-in-a-Dish models” [84]. NEC-in-a-Dish organoids reproduce features consistent with human organoids derived from NEC-affected tissue, such as increased TLR4 expression, cell proliferation, tight junction disruption, apoptosis, and impaired Wnt/β-catenin signaling [84]. Importantly, TLR4 inhibitors and human milk restore proliferation in LPS-treated organoids, confirming their appropriateness for therapeutic testing.

**Gut-on-a-chip model system:** Gut-on-a-chip model system represents a new class of microengineered models that incorporate flow, stretch, and co-cultures with immune, vascular, and microbial cells [102,103]. These platforms enable real-time monitoring of epithelial barrier function, immune signaling, and microbial interactions. For example, LPS exposure on gut chips induces IL-6, IL-8, and TNF-α secretion and disrupts tight junctions—hallmarks of NEC. Co-culturing with commensals and Peripheral Blood Mononuclear Cells (PBMCs) enables the modeling of inflammation and barrier rescue.

**Microfluidic NEC-on-a-Chip model:** A recent advancement in in vitro modeling is the development of a microfluidic NEC-on-a-Chip platform that co-cultures human neonatal enteroids with intestinal microvascular endothelial cells under dynamic flow conditions, which offers a scalable and patient-specific tool for mechanistic studies and preclinical therapeutic testing in NEC [86,102,103]. Upon exposure to NEC-associated microbiota, this model recapitulates hallmark features of NEC, including epithelial barrier disruption, inflammatory cytokine upregulation, loss of stem and secretory cells, and activation of cell death pathways.

**Human Explant NEC models:** The human explant tissues (ex vivo modeling) derived from de-identified post-surgical resected specimens, which are treated transiently with stressors and therapeutic agents for 6 to 8 h in culture, have proven to be excellent tools for studying NEC pathogenesis [18,97]. Together, in vitro platforms are improving our understanding of NEC at both cellular and molecular levels. As tools for examining host-microbe interactions, barrier dysfunction, immune regulation, and therapeutic testing, these models are crucial for translating fundamental discoveries into clinical interventions.

**Choosing the Right Model:** Although no model perfectly replicates human NEC, each one has significantly enhanced our understanding of the disease. The selection of an NEC model should be based on the biological question being asked: genetic and molecular mechanisms are best addressed in mouse models; therapeutic interventions and enteral nutrition in large animal models with genetic and physiological proximity to humans, and cellular signaling and host-microbe interactions in organoids and ex vivo systems. Continued development of integrative and multi-scale NEC models—linking epithelial, microbial, immune, and vascular components—will be critical to translating bench findings into clinical interventions.

### 2.2. Compromised Gut Epithelium Triggering NEC Pathogenesis

**Immature Intestinal Barrier.** The intestinal barrier forms the first line of defense between the host and the luminal environment. In preterm infants, this barrier is structurally and functionally immature, rendering it highly susceptible to microbial translocation and inflammation, a hallmark of NEC [105]. Disruption of the intestinal barrier facilitates the entry of opportunistic pathogens, microbial products, and toxins into the mucosa and systemic circulation, triggering a cascade of immune activation, oxidative epithelial injury, apoptosis, necroptosis, and necrosis. The intestinal epithelium in premature neonates exhibits underdeveloped tight junctions, crucial for maintaining selective permeability to nutrients and preventing the invasion of pathogens [105,106]. This immaturity, in combination with mucosal stresses such as hypoxia, allows luminal bacteria and endotoxins, such as LPS, to penetrate the epithelium and activate TLR4, leading to inflammation and tissue injury [20,21]. Several studies have shown that tight junction proteins (e.g., claudins, occludin, ZO-1) are significantly downregulated or mis-localized in NEC [18,21].

**Goblet Cell Dysfunction and Mucin Deficiency.** Goblet cells, which secrete glycoproteins called mucins to form the mucus layer, are decreased in the intestines of both human infants and mouse models with NEC [96,107]. Mucins protect the epithelium by lubricating the surface and preventing direct microbial contact. Mucin deficiency has been linked to increased microbial adherence and disruption of the barrier. Pharmacologic activation of goblet cell differentiation using Dibenzazepine, a γ-secretase inhibitor of Notch signaling, has been shown to increase goblet cell numbers and provide protection against NEC [96].

**Paneth Cell Deficiency and Antimicrobial Impairment.** Paneth cells are specialized long-lived epithelial cells (renewed every 3–4 weeks), which reside at the bottom of the intestinal epithelium, i.e., crypts of Lieberkühn, and produce antimicrobial peptides (AMPs), including defensins and lysozyme, which are essential for maintaining microbial balance and preventing pathogenic overgrowth [108]. In preterm infants, Paneth cells are either absent or functionally immature, resulting in a deficiency in antimicrobial defense. Studies have shown that lysozyme-positive Paneth cells are reduced in NEC-affected infants compared to age-matched surgical controls, indicating that either degranulation or cell loss occurs before NEC develops [109,110].

**Impaired Epithelial Regeneration and Healing.** Epithelial regeneration is a crucial process that preserves the integrity and functionality of the mucosal epithelial barrier, especially after chemical or physical injury. Under normal conditions, intestinal stem cells (ISCs) residing at the bottom of the crypts of Lieberkühn proliferate and produce absorptive enterocytes and secretory lineages (goblet, Paneth, and enteroendocrine cells). These cells migrate upward along the crypt-villus axis, undergo final differentiation, and are shed at the tips, allowing complete epithelial renewal every 3–5 days. This coordinated cycle preserves mucosal homeostasis and defends against luminal insults [111]. In NEC, this regenerative process becomes severely disrupted. Increased activation of TLR4 in the premature intestine hampers key regenerative pathways by suppressing β-catenin signaling and downregulating Lgr5, a primary ISC marker [112,113,114]. As a result, stem cell proliferation decreases, and epithelial migration toward the injury site slows. Additionally, TLR4 activation stimulates endoplasmic reticulum (ER) stress, leading to apoptosis of crypt-epithelial cells, further depleting the regenerative pool [112]. The overall effect is a significant delay in mucosal healing, impaired wound repair, and ongoing exposure to microbial products, further producing the inflammatory insult. This faulty repair mechanism exacerbates barrier dysfunction and perpetuates the cycle of inflammation and injury characteristic of NEC [18,112,115].

**Microbial Dysbiosis.** The preterm intestine is uniquely vulnerable to microbial dysbiosis, a state of disturbed microbial composition and function that plays a central role in the pathogenesis of NEC. Rather than being caused by a single pathogen, NEC arises from an abnormal microbial colonization pattern that triggers dysregulated host-microbe interactions in an immunologically immature gut [116]. Studies have shown that NEC is preceded by (1) a loss of microbial diversity, (2) an overgrowth of Gram-negative Proteobacteria, particularly Enterobacteriaceae, (3) a deficiency in beneficial commensals, including Bifidobacterium and Lactobacillus, and (4) a failure of colonization by Firmicutes and Bacteroidetes, which are typically present in healthy term infants. These changes occur early—often days before NEC diagnosis—and are accentuated in formula-fed or antibiotic-exposed neonates. Dysbiosis promotes mucosal injury through several mechanisms: TLR4 activation by Gram-negative LPS [114,115], depletion of short-chain fatty acids (SCFAs) like butyrate [91], alterations in the mucus barrier [96,107], and enzymatic degradation of sialylated glycans by microbial Sialidases [24]. Pathogenic bacteria, such as Clostridium perfringens, secrete sialidases that remove protective terminal sialic acids from mucins and epithelial glycoproteins, thereby exposing the underlying TLR4-rich epithelium to inflammatory triggers. These enzymes also liberate free sialic acid, favoring further pathogenic growth and tipping the microbial balance [24]. The 2024 study by Klerk et al. demonstrated that multi-strain probiotic supplementation (*Bifidobacterium infantis*, *Bifidobacterium lactis*, and *Streptococcus thermophilus*) restored microbial balance and reduced NEC severity in a murine ileo-colitis model [23]. These effects included the suppression of pro-inflammatory cytokines (e.g., TNF-α, IL-1β), enhanced gut motility, and reduced nuclear translocation of nuclear factor-kappa B (NF-κB). Significantly, probiotics also reversed NEC-associated DNA methylation changes at key gene loci related to inflammation and barrier regulation (e.g., Tacr3, Tiam1, HoxA1). Even under non-NEC conditions, probiotic supplementation induced significant epigenetic changes in intestinal tissue, suggesting that probiotics can prime the mucosal immune system for resilience. These data highlight that probiotics serve not only as immunomodulators but also as epigenetic regulators of host-microbe interactions [23]. Future research should investigate probiotic strain-specific activity against sialidase-producing pathogens, explore synergistic interactions with human milk oligosaccharides, and examine the potential for epigenomic-guided probiotic personalization in high-risk neonates.

### 2.3. Toll-like Receptor 4 (TLR4) as a Key Driver of NEC Pathogenesis

**TLR family:** Among the family of Toll-like receptors (TLRs), TLR4 has a uniquely central role in driving the inflammation associated with NEC [20,112,113,114,115] (Figure 1 and Figure 2). While other TLRs—such as TLR2, TLR5, and TLR9—are involved in detecting microbial products and managing mucosal immunity, their activation in the immature intestine is usually linked to less severe epithelial damage or even protective effects in certain cases [117]. Unlike TLR4, TLR2 detects components of Gram-positive bacteria such as lipoteichoic acid and peptidoglycan [118]. In the immature gut, its activation can strengthen the epithelial barrier by increasing tight junction proteins [119]. Experimental models show that TLR2 stimulation may reduce TLR4-driven inflammation, suggesting a potential modulatory role in NEC development [120]. TLR5 detects bacterial flagellin and is found on intestinal epithelial cells during early life. While its activation can boost protective immune responses and influence microbiota composition, abnormal TLR5 signaling in the developing intestine may lead to inflammation, though its role in NEC is less understood than TLR4 or TLR2 [121]. Finally, TLR9 recognizes unmethylated CpG motifs in bacterial DNA and is expressed in the intestinal epithelium and immune cells [122]. In NEC models, TLR9 activation has been shown to downregulate TLR4 signaling and reduce intestinal inflammation, suggesting a potential protective role in maintaining mucosal homeostasis in the premature gut [123].

**TLR4 expression and signaling in the immature gut.** TLR4, a receptor for bacterial LPS, is expressed in various cell types—including intestinal epithelium [96,112], immune cells [21], enteric nervous system cells [124], and vascular endothelium [125]—expressions reported based on cell-specific TLR4 knockout techniques, since dependable TLR4 antibodies are nonexistent. While TLR4 plays a protective role in the mature host by triggering defensive responses to pathogens, its activation leads to detrimental outcomes in the immature intestine. In the developing intestine, TLR4 expression is markedly upregulated, particularly in the epithelial cells of the ileum—the region most affected in NEC [21,96,123]. This developmental overexpression is believed to reflect the receptor’s role in regulating intestinal maturation [96]. However, it also predisposes the premature gut to an exaggerated response to microbial stimuli. Upon exposure to LPS from Gram-negative bacteria, TLR4 initiates a MyD88-dependent signaling cascade, culminating in the activation of nuclear factor kappa B (NF-κB) and the release of potent pro-inflammatory cytokines, including TNF-α, IL-6, and IL-1β. These cytokines recruit and activate immune cells (e.g., macrophages, neutrophils), amplifying the inflammatory milieu and contributing to epithelial damage [21].

**Aberrant TLR4 signaling in intestinal stem cells causes NEC.** Emerging evidence shows that TLR4 is expressed not only in differentiated epithelial cells but also in Lgr5^+^ intestinal stem cells (ISCs) located at the crypt base [20,112,113]. In the immature intestine, abnormal activation of TLR4 in ISCs by bacterial LPS suppresses the Wnt/β-catenin signaling pathway, which is vital for ISC proliferation, stem cell maintenance, and epithelial regeneration. This suppression results in reduced ISC renewal, progressive loss of stem cells, and impaired mucosal healing after injury—a defect especially harmful during the rapid growth and repair phase of the neonatal period. In experimental NEC, targeted inhibition or conditional deletion of TLR4 in ISCs preserves stem cell function, speeds up epithelial recovery, and significantly decreases disease severity [20,112,113]. Mechanistically, TLR4 activation in ISCs causes endoplasmic reticulum (ER) stress via the PERK–CHOP axis, leading to crypt-specific apoptosis [112]. This ER stress is inherently high in the immature intestine even at baseline, priming the premature gut for heightened injury upon microbial colonization. Genetic removal of PERK or CHOP in intestinal crypts offers significant protection against NEC-related epithelial loss and inflammation. Additionally, the pro-apoptotic protein PUMA (*p53 upregulated modulator of apoptosis*) has been identified as a downstream effector connecting ER stress to mitochondrial apoptosis. Blocking PUMA prevents crypt cell apoptosis and reduces NEC severity, suggesting that TLR4–PERK–CHOP–PUMA signaling is a key pathway through which ER stress links microbial sensing to ISC loss and mucosal damage. Overall, these findings establish aberrant TLR4 signaling in ISCs as an essential link between innate immune activation, defective epithelial regeneration, and NEC development. They also highlight ISC-protective strategies as an encouraging therapeutic option.

**TLR4 activation causes Ischemia and Impaired Blood Flow in NEC.** Ischemia, defined as the inadequate delivery of blood to tissues, plays a critical and underappreciated role in the pathogenesis of NEC [85,126]. In the context of NEC, impaired perfusion of the intestinal microvasculature leads to tissue hypoxia, energy depletion, mucosal injury, and necrosis. This vulnerability is particularly pronounced in the premature intestine, where vascular autoregulation and oxygen delivery mechanisms are underdeveloped. Although the precise etiology of ischemia in NEC remains a topic of debate, it is widely acknowledged as a compounding factor linked to prematurity, formula feeding, and inflammation. Yazji et al. conducted elegant studies using fluorescein-labeled tomato lectin to assess intestinal microvascular perfusion in a neonatal mouse model of NEC [125]. Their findings revealed that TLR4 activation on endothelial cells led to significant hypoperfusion of the intestinal microcirculation. This vascular dysfunction contributed to mucosal injury and disease progression. Importantly, genetic deletion of TLR4 specifically in endothelial cells restored perfusion and protected mice from NEC, implicating endothelial TLR4 as a direct driver of ischemia. Several interventions have been shown to restore intestinal perfusion and ameliorate NEC severity: vasodilatory agents, including Sildenafil (a phosphodiesterase-5 inhibitor) and exogenous nitrate supplementation, improved microvascular flow when administered in neonatal formula, thereby reducing NEC incidence [125]. Human milk oligosaccharides (HMOs), known for their immunomodulatory effects, were shown by Good et al. [26] to also restore perfusion in NEC models, suggesting that breast milk may protect against NEC in part by preserving vascular integrity. Senarathna et al. [127] applied multi-contrast optical imaging using intravenous ultrasmall gold nanoclusters and demonstrated real-time intestinal hypoperfusion in NEC mice, confirming that reduced vascular flow precedes overt histologic injury. In a novel approach, Jones et al. applied remote ischemic conditioning (RIC) in a rat model of NEC and conducted transcriptomic analyses to examine downstream effects [128]. RIC conferred intestinal protection through anti-inflammatory signaling, reducing expression of genes involved in oxidative stress and inflammation, including NF-κB2, Cxcl1, SOD2, and Map3k8. These findings suggest that modulating blood flow and vascular signaling can influence not only perfusion but also epithelial and immune gene expression, highlighting the cross-talk between vascular and inflammatory pathways in NEC.

**Aberrant TLR4 signaling causes oxidative and mucosal injury in NEC (Figure 2).** The immature intestine naturally lacks key endogenous antioxidant defenses—such as superoxide dismutase (SOD), catalase, and glutathione peroxidase—making it highly vulnerable to oxidative damage [129,130]. Under NEC conditions, which involve hypoxia–reoxygenation injury, bacterial translocation, and intense mucosal inflammation, both intestinal epithelial cells and infiltrating immune cells—particularly activated neutrophils and macrophages—produce large amounts of reactive oxygen species (ROS) and reactive nitrogen species (RNS), including nitric oxide (NO) [27,131]. These reactive intermediates promote lipid peroxidation of cell membranes, cause oxidative damage to DNA and proteins, and disrupt mitochondrial respiration, all leading to epithelial cell death and barrier dysfunction [101]. TLR4 activation significantly increases oxidative stress in NEC [27,131]. When TLR4 is stimulated by bacterial lipopolysaccharide (LPS), it triggers NADPH oxidase (NOX) activation in both epithelial and immune cells, boosting ROS production [18,132]. This process is accompanied by the upregulation of inducible nitric oxide synthase (iNOS), which raises NO levels and can react with superoxide to produce peroxynitrite—a highly cytotoxic oxidant involved in epithelial injury [133]. ROS and RNS activate pro-inflammatory transcription factors such as NF-κB and AP-1, further amplifying cytokine release and inflammatory cascades. The oxidative injury from TLR4-mediated signaling also disturbs the epithelial redox balance, leading to the disassembly of tight junctions through oxidation of junctional proteins like occludin, claudins, and ZO-1. This barrier breakdown increases paracellular permeability, promoting bacterial translocation and perpetuating the cycle of inflammation and oxidative damage [134].

**Aberrant TLR4 signaling leading to Necroptosis and inflammatory cell death in NEC (Figure 3).** Necroptosis is a programmed, caspase-independent form of cell death that culminates in plasma membrane rupture, release of damage-associated molecular patterns (DAMPs), and robust activation of innate immune responses [135]. Unlike apoptosis, which is immunologically silent, necroptosis actively propagates inflammation and has emerged as a key contributor to the pathogenesis of NEC [84]. This form of cell death is mediated by sequential activation of receptor-interacting protein kinase 1 (RIPK1), RIPK3, and mixed lineage kinase domain-like protein (MLKL), which translocate to the membrane to induce pore formation and cellular lysis. In experimental NEC models, RIPK3 is markedly upregulated in the ileum, particularly within the villus epithelium, where necroptosis appears to dominate over apoptotic mechanisms [84]. In contrast to the crypt-localized apoptosis that targets intestinal stem cells, necroptosis primarily affects the differentiated epithelial layer, leading to widespread mucosal sloughing and exposure of the lamina propria [112]. Necroptosis also promotes TLR4-mediated inflammation by releasing DAMPs such as HMGB1 and mitochondrial DNA into the lumen and systemic circulation, further stimulating macrophage and neutrophil recruitment [136]. These findings position necroptosis not only as a driver of tissue damage but also as a key link between epithelial injury and systemic immune activation in NEC. Targeting the necroptotic machinery represents a promising therapeutic strategy, particularly in combination with agents that preserve stem cell viability and restore mucosal regeneration.

**Non-TLRs in Influencing NEC Pathogenesis:** Along with TLR4, other non-TLR innate immune pathways are involved in NEC development, offering a broader view of the host’s response to microbial and injury signals. Nucleotide-binding oligomerization domain-like receptors, such as Nucleotide-binding oligomerization domain 2 (NOD2), detect bacterial peptidoglycan motifs and can influence inflammatory responses in the gut [137]. Loss-of-function variants of NOD2 have been linked to increased intestinal inflammation and compromised barrier integrity [138]. Additionally, inflammasomes—multi-protein complexes that activate caspase-1—are important for processing pro-inflammatory cytokines such as IL-1β and IL-18 [137]. In NEC models, abnormal activation of the NLRP3 inflammasome has been associated with epithelial cell death and severe mucosal damage [139]. Collectively, these pathways interact with TLR4 signaling to form the complex innate immune response that drives NEC onset and progression.

### 2.4. Compromised Immune Landscape in NEC Pathogenesis

Myeloid cells are key components of the innate immune system and include monocytes, macrophages, dendritic cells, and granulocytes such as neutrophils, basophils, eosinophils, and mast cells. These cells provide a rapid, nonspecific defense against pathogens. In contrast, the adaptive immune system—comprising T and B lymphocytes—offers a highly specific response, targeting distinct pathogens and antigens through antigen-specific receptors [140]. NEC involves an exaggerated and dysregulated immune response in the immature intestine. Innate immune cells contribute to mucosal injury through the release of pro-inflammatory cytokines and the formation of neutrophil extracellular traps (NETs), amplifying local inflammation and tissue damage [141,142,143,144]. Moreover, an imbalance between pro-inflammatory Th17 cells and immunosuppressive regulatory T cells (Tregs) exacerbates intestinal inflammation, partly driven by TLR4–STAT3 signaling, a critical upstream pathway in NEC pathogenesis [21]. Emerging evidence also implicates the involvement of Th2-associated cells, including eosinophils and mast cells, in the inflammatory cascade of NEC, suggesting a broader immune dysregulation beyond classical Th1/Th17 pathways [145,146]. Collectively, this aberrant immune environment contributes to intestinal barrier breakdown, mucosal necrosis, and systemic inflammation—the hallmarks of NEC [147].

**Monocytes and Macrophages.** Monocytes and macrophages are versatile immune cells involved in tissue homeostasis, injury response, and repair, exhibiting high functional and phenotypic heterogeneity depending on disease context and tissue environment. In the intestine, macrophages can be broadly categorized into lamina propria macrophages and muscularis macrophages, each occupying distinct niches and engaging in specialized interactions with surrounding structures such as the vasculature and the enteric nervous system (ENS). During embryogenesis, intestinal macrophages originate from yolk sac-derived primitive macrophages (pre-macrophages) and/or fetal liver-derived precursors. Yolk sac macrophages rapidly proliferate and migrate to target tissues, where they diversify upon arrival [148]. As fetal liver hematopoiesis develops and systemic circulation is established, macrophages derived from this later wave of hematopoiesis also colonize the intestine and contribute to the pool of tissue-resident macrophages [149,150]. In the postnatal period, intestinal macrophages gradually shift toward a population derived from bone marrow (BM)-origin hematopoietic stem cells, with BM-derived monocyte precursors becoming the dominant source in adulthood. Nevertheless, a subset of long-lived, self-renewing resident macrophages persists, particularly near the submucosal and myenteric plexuses, where they are in close proximity to vasculature and ENS elements [150,151,152]. In neonates, resident macrophages are the predominant population, but over time, circulating monocytes progressively replace them through conventional hematopoiesis [153]. In adult models of intestinal disease, studies have shown that depleting lamina propria macrophages or blocking BM-derived macrophage infiltration can reduce mucosal inflammation and injury in various colonic disorders [154,155,156,157,158,159]. Similarly, in NEC models, depletion of monocyte-derived macrophages or inhibition of their recruitment attenuates disease severity, underscoring their pathogenic role [160]. However, the functional role of resident muscularis macrophages in NEC remains poorly defined. While these cells are known to regulate intestinal motility and secretion, their contribution to NEC pathogenesis—especially their potential activation and inflammatory crosstalk with the ENS—requires further investigation. Importantly, the immunological phenotype of intestinal macrophages differs significantly between full-term and preterm neonates. Full-term infants exhibit minimal inflammatory cytokine responses to microbial stimuli and display enhanced phagocytic and bactericidal functions. In contrast, preterm neonates exhibit a hyperinflammatory macrophage profile, with increased production of cytokines such as IL-1β, IL-6, TNF-α, and chemokines by CD16^+^CD163^+^ monocyte-derived macrophages [161,162,163]. This heightened inflammatory state contributes to mucosal injury and barrier dysfunction. Mechanistically, TLR4-mediated signaling in immature intestinal macrophages plays a central role in amplifying inflammatory responses to microbial colonization. This pathway promotes excessive cytokine production, mucosal injury, and systemic inflammation—hallmarks of NEC pathogenesis [21,142,164].

**T Cells.** T cells play a critical and mechanistic role in the pathogenesis of NEC. Both human and murine NEC ileal tissues show abundant T cell infiltration, and their presence is essential for disease development. T cell–deficient mice (Rag knockout) are resistant to NEC, while the adoptive transfer of T cells from NEC mice into naïve recipients induces intestinal inflammation, demonstrating their pathogenic potential [21]. The recruitment of T cells into the inflamed intestine during NEC is TLR4-dependent, mediated through the CCR9–CCL25 chemokine axis and STAT signaling pathways [21]. Beyond the gut, NEC-associated gut-derived CD4^+^ T cells can migrate to the brain, where they trigger neuroinflammation and brain injury via interferon-γ (IFN-γ)–mediated mechanisms [8]. In human NEC ileal tissues, the ratio of regulatory T cells (Tregs; FoxP3^+^) to effector T cells (CD4^+^ and CD8^+^) is significantly reduced compared to controls. This imbalance is associated with increased expression of pro-inflammatory cytokines [165,166]. Experimental NEC models similarly demonstrate a marked decrease in intestinal Tregs and a concurrent increase in Th17 cells, characterized by expression of RORγt and production of IL-17. Adoptive transfer of Tregs into these models significantly attenuates disease severity and improves survival, further supporting the regulatory role of Tregs in limiting NEC-associated inflammation [21,165]. These findings collectively highlight the pathogenic importance of a Th17/Treg imbalance in NEC. An environment characterized by elevated IL-17 production and diminished Treg-mediated immunoregulation contributes to intestinal epithelial damage, mucosal inflammation, and systemic complications. In addition to conventional T cells, intraepithelial γδ T cells have also been implicated in NEC. These cells are significantly depleted in the ileum of NEC patients, and their reduction is associated with elevated IL-17 levels, further supporting their potential protective role in maintaining mucosal immune balance [164,166].

**Neutrophils.** Neutrophils are among the earliest responders in the pathogenesis of NEC and play a dual role in both mucosal defense and tissue injury. Neutrophil infiltration is commonly observed in NEC lesions and is believed to contribute to the disruption of mucosal homeostasis [144]. Experimental NEC models induced by *Cronobacter sakazakii* demonstrate that neutrophil depletion impairs bacterial clearance, increases enterocyte apoptosis, and elevates cytokine release, suggesting a protective role for neutrophils in host defense [167]. However, excessive neutrophil activation can be detrimental: neutrophil-derived reactive oxygen species (ROS) and platelet-activating factor (PAF) contribute to epithelial injury and gut inflammation [168]. A more recent study using Elastase, Neutrophil Expressed (ELANE) knockout mice, which lack the ability to produce neutrophil elastase—a proteolytic enzyme involved in the breakdown of proteins—showed that inhibition of neutrophil elastase protects the ileum from NEC-induced damage [169]. A growing body of evidence implicates neutrophil extracellular traps (NETs)—web-like DNA structures released by activated neutrophils to trap and kill microbes—as important mediators in NEC pathogenesis. NET formation has been observed in NEC tissues and correlates with intestinal injury [170,171]. Importantly, inhibition of NETs using protein arginine deiminase (PAD) inhibitors significantly improves survival, reduces mucosal damage, and dampens intestinal inflammation in NEC models, highlighting NETs as a potential therapeutic target. In human NEC tissues, single-cell RNA sequencing has identified increased populations of immature, newly emigrated, and aged neutrophil subsets, which correlate with elevated levels of IL-6 and IL-8—key inflammatory chemokines in NEC [163]. Additionally, the presence of CD56^+^ neutrophils in NEC mucosa has been positively correlated with CD16^+^CD163^+^ pro-inflammatory macrophages, suggesting dynamic cross-talk between neutrophils and macrophages in amplifying the inflammatory milieu of NEC [142]. Collectively, these findings underscore the complex, context-dependent role of neutrophils in NEC—serving as both defenders against microbial invasion and potential amplifiers of mucosal injury when excessively activated.

**Mast Cells and Eosinophils.** Although eosinophils and mast cells are traditionally recognized for their roles in allergic responses and type 2 immunity, emerging evidence suggests they may also contribute to the pathogenesis of NEC [66]. Eosinophil infiltration in the gastrointestinal wall has been observed in infants with NEC and bloody stools, though the functional significance of this finding remains incompletely understood [66,146,172]. A recent systematic review proposed a dual role for eosinophils in NEC—functioning as inflammatory mediators during disease onset and as potential contributors to tissue repair during the resolution phase [17]. Eosinophils can release cytotoxic granule proteins, such as eosinophil peroxidase and major basic protein, which cause direct epithelial injury and can also stimulate mast cell degranulation, thereby amplifying the inflammatory response [173]. Mast cells, when activated by Th2-type cytokines (e.g., IL-4, IL-5, IL-13) or eosinophil-derived granules, release a range of pro-inflammatory mediators including histamine, proteases, and lipid-derived mediators. These substances promote increased vascular permeability, facilitate immune cell infiltration, and contribute to local tissue damage [174]. Although the precise roles of eosinophils and mast cells in NEC pathogenesis remain to be fully elucidated, current evidence points to a potentially underrecognized eosinophil–mast cell axis that could reflect a Th2-skewed immune component in NEC. This axis may contribute to both the propagation and resolution of inflammation, suggesting a more complex immunopathology than previously appreciated.

### 2.5. Compromised Enteric Nervous System in NEC Pathogenesis

The enteric nervous system (ENS) plays a critical role in regulating intestinal motility, epithelial barrier integrity, and immune responses [175]—functions that are profoundly disrupted in NEC [124,176]. Emerging studies have identified ENS injury as a major contributor to NEC pathogenesis, adding a neuroimmune dimension to what was previously considered primarily an inflammatory disease of the epithelium. In a Pivotal study by Kovler et al., NEC was shown to cause a marked loss of enteric neurons and glial cells in the distal ileum [124]. This neuronal injury was accompanied by inflammation-induced suppression of neurotrophic factors including glial cell line-derived neurotrophic factor (GDNF) and brain-derived neurotrophic factor (BDNF). The reduction in these protective signals coincided with downregulation of Sox10, and other genes critical for neuronal maintenance and synaptic function. These findings were corroborated in both mouse models and human NEC intestinal samples. Moreover, the study identified a small-molecule compound, agent J1, that activates the TrkB receptor (the cognate receptor for BDNF). Treatment with J1 preserved neuronal and glial populations, reduced intestinal inflammation, and improved survival in experimental NEC. Notably, BDNF/TrkB signaling restored both neural and epithelial integrity, suggesting cross-talk between the ENS and mucosal injury pathways. These data suggest that ENS damage in NEC is not simply a consequence of inflammation but a driver of disease severity. Restoration of BDNF-TrkB signaling and pharmacologic protection of enteric neurons and glia via agent J1 offer promising therapeutic strategies for NEC, targeting the often-overlooked neuroimmune axis of this devastating disease. Together, these mechanisms form a unifying hypothesis: NEC arises due to a TLR4-driven imbalance between epithelial injury and inadequate repair in the context of microbial signals and vascular /neuronal compromise.

### 2.6. Translational Advances: NEC Prevention and Treatment

**Probiotics:** Probiotics reduce NEC incidence and severity, especially with strains of Bifidobacterium and Lactobacillus [22,23,177]. Mechanisms include strengthening epithelial barrier function, preventing pathogen colonization, modulating innate immune responses, and promoting beneficial microbial metabolites. Multi-strain and targeted probiotic formulations, such as those used in Klerk et al., have shown benefits in experimental NEC through anti-inflammatory, motility-enhancing, and even epigenetic reprogramming effects [23]. However, variability in clinical effectiveness and regulatory issues limit widespread use. Ongoing trials aim to improve strain selection, dosage, and timing.

**Pharmacological and Molecular Interventions:** Mechanistic studies have identified several pathways suitable for pharmacological targeting: TLR4 inhibition through small molecules or genetic suppression [19,96,112]; antioxidants like N-acetylcysteine [7,27] and melatonin to decrease oxidative stress [178]; RIPK3 and NLRP3 inhibitors to prevent necroptosis and pyroptosis [84,135,136]; BDNF-mimetic peptides such as Agent J1, which restore enteric neuronal networks and improve intestinal motility [124]; and Notch signaling inhibitors (e.g., dibenzazepine) to increase goblet cell populations [96]. These agents have demonstrated preclinical efficacy, although few have moved to neonatal trials due to safety and delivery issues.

**Stem Cell and Regenerative Therapies:** Recent studies indicate that mesenchymal stem cells (MSCs) [179], Amniotic fluid stem cells (AFSCs) [180], and intestinal stem cell-derived vesicles [181] may reduce NEC by releasing anti-inflammatory and angiogenic factors and supporting epithelial regeneration. Maltais-Bilodeau et al.’s recent systematic review and meta-analysis highlight cell-based therapies in preclinical models of NEC [182].

**Bioengineering and Translational Platforms:** Organ-on-a-chip technologies, including gut-on-a-chip and NEC-on-a-chip, provide innovative platforms for modeling host–microbe–immune interactions and screening therapeutics under dynamic conditions. These systems simulate intestinal peristalsis, flow, and cell co-culture environments; allow testing of enteral or systemic agents; and offer insights into epithelial–endothelial–immune cross-talk. These platforms supplement preclinical models and help bridge the gap between discovery and clinical translation [86,102]. All translational advances are summarized in Table 3.

## 3. Future Directions and Conclusions

NEC remains a significant clinical challenge in neonatology, with multifactorial causes and limited treatment options. As outlined in the Introduction, advances in clinical, immunological, microbial, and pathophysiological research have improved our understanding of disease mechanisms, yet major gaps remain in early detection, risk assessment, and effective interventions. To close these gaps, progress will depend on a multidisciplinary framework that bridges basic science, translational research, and clinical application.

**Precision Neonatology and Risk Prediction.** Prospective diagnostic approaches are shifting toward precision medicine, where integration of clinical variables with microbial, genetic, and epigenetic data may allow NEC to be predicted before clinical onset. Emerging tools such as cytokine panels, fecal and urinary metabolomics [183], methylation signatures [23,44], and real-time microbiome sequencing [184], combined with machine learning models trained on large NICU datasets [31,32], show promise for personalized risk assessment and early intervention. These advances could enable tailored feeding strategies, targeted use of probiotics or anti-inflammatory therapies, and dynamic surveillance.

**Mechanism-Guided Therapies:** Moving beyond supportive care, future NEC therapies must target fundamental drivers such as TLR4 and inflammasome activation. Oxidative stress has already shown the efficacy of TLR4 inhibitors [19,114], antioxidant nanoparticles [7,130], exosome-based therapies [185], and modulators of enteric nervous system function [124]. Translating these into neonatal settings will require rigorous testing in small- and large-animal models and carefully designed early-phase clinical trials.

**Microbiome Engineering and Postbiotics:** Microbiota-based strategies are quickly expanding beyond traditional probiotics. These include intentionally designed microbial consortia, engineered probiotics with specific metabolic features, postbiotics like short-chain fatty acids or indole derivatives that help restore epithelial balance, and bacteriophage therapy to accurately target pathobionts [177,185]. Considering host genotype, diet, and microbial function will be essential for developing personalized microbial therapies to prevent NEC.

**Regenerative and Stem Cell-Based Interventions:** Regenerative medicine offers promising solutions for NEC survivors with short bowel syndrome or chronic inflammation. Amniotic fluid and mesenchymal stromal cells reduce intestinal injury, promote angiogenesis, and modulate inflammation [180,182]. Future strategies include intestinal organoid transplantation and tissue-engineered scaffolds for mucosal repair and restoration [86]. Challenges involve optimizing engraftment, avoiding rejection, and scalable production.

**Multisystem and Long-Term Sequelae:** NEC is increasingly recognized as a systemic disease, with long-term effects on the brain, lungs, and metabolism. Future research should explore the gut–brain axis, including neuroinflammation and cognitive outcomes [7]; pulmonary injury caused by systemic inflammation and shared vascular signaling [6]; and growth and developmental paths of NEC survivors into childhood. Longitudinal cohorts and multi-omics profiling will be crucial to fully understand NEC’s lasting effects.

**Strategies to Prevent and Delay Preterm Birth:** NEC is primarily a disease of prematurity, and therefore a preventative approach should include strategies to prevent or delay pregnancies at risk for preterm delivery. According to practice guidelines from the American College of Obstetricians and Gynecologists, key strategies include maternal smoking and alcohol cessation, improvements to maternal nutrition, increasing interpregnancy intervals to at least 18 months, and providing access to contraception [186]; targeted public health policies and interventions to evaluate their success are also necessary. Additionally, further investigation is needed into various emerging therapies for preventing spontaneous preterm birth, including l-arginine, selenium, and lactoferrin, as well as new tocolytics and anti-inflammatory drugs [187].

## 4. Concluding Remarks

NEC disease highlights how immature physiology, microbial imbalance, and hyperinflammation are interconnected and influenced by timing, nutrition, and genetics. Progress depends on collaboration across fields such as neonatology, surgery, immunology, microbiology, bioengineering, and data science to mitigate the disease’s effects. Understanding the underlying mechanisms and combining recent technological advances with patient-focused treatments (such as stem cell therapy) are key to developing treatment strategies. By implementing these strategies, NEC can evolve from a mainly reactive emergency to a preventable and manageable condition with precision medicine. This positive change could greatly enhance outcomes for our most vulnerable infants.

## Figures and Tables

**Figure 1 biomedicines-13-02288-f001:**
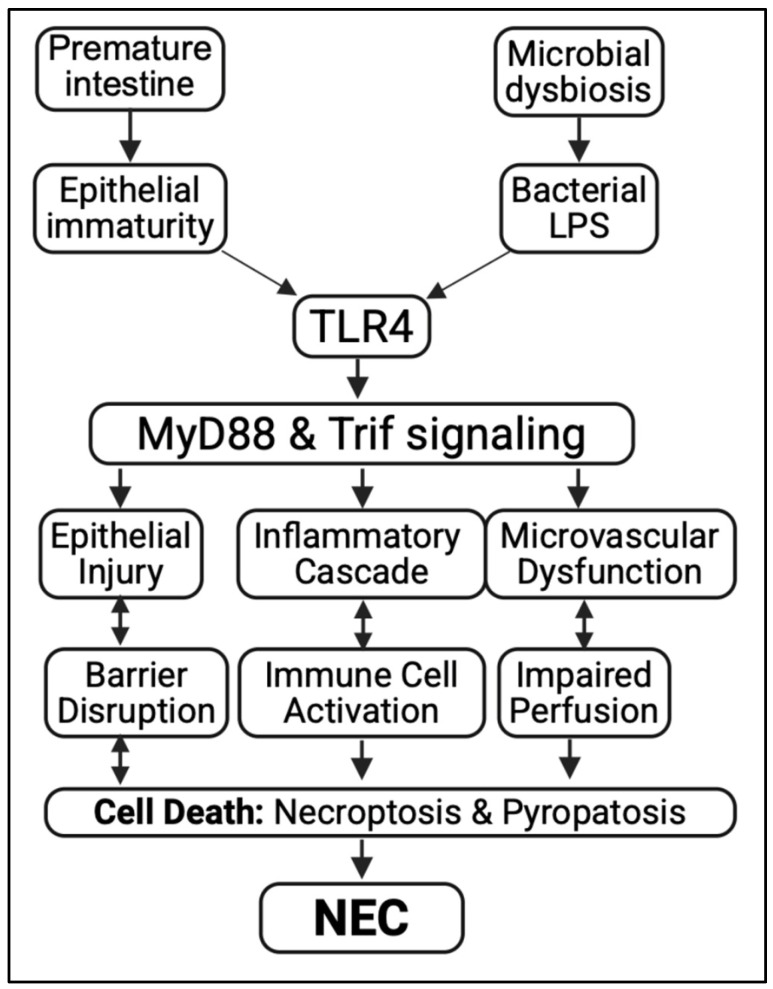
**TLR4-mediated signaling in the pathogenesis of NEC.** Prematurity and microbial dysbiosis activate TLR4 through bacterial LPS, triggering MyD88/Trif signaling. This causes epithelial injury, inflammation, and microvascular dysfunction, leading to cell death and the development of necrotizing enterocolitis (NEC).

**Figure 2 biomedicines-13-02288-f002:**
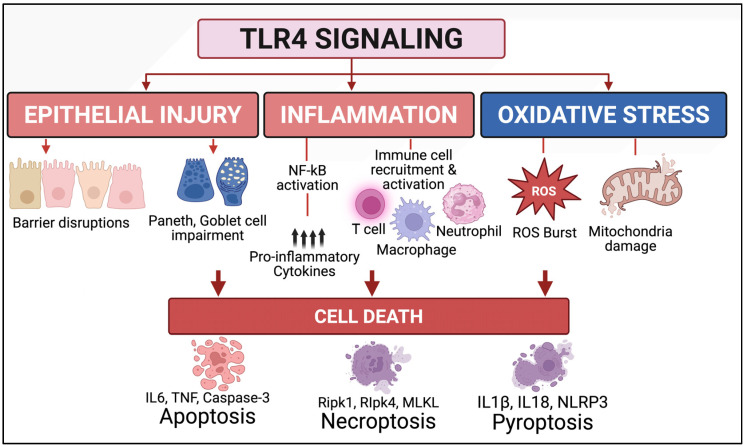
**TLR4 signaling drives multiple regulated cell death pathways in NEC.** TLR4 activation in the premature intestine triggers epithelial injury (barrier disruption, Paneth and goblet cell impairment), inflammation (NF-κB activation, cytokine release, immune cell recruitment), and oxidative stress (ROS generation, mitochondrial damage). These processes collectively activate various programmed cell death pathways: apoptosis (IL-6, TNF, caspase-3), necroptosis (RIPK1, RIPK3, MLKL), and pyroptosis (IL-1β, IL-18, NLRP3), worsening intestinal injury in NEC.

**Figure 3 biomedicines-13-02288-f003:**
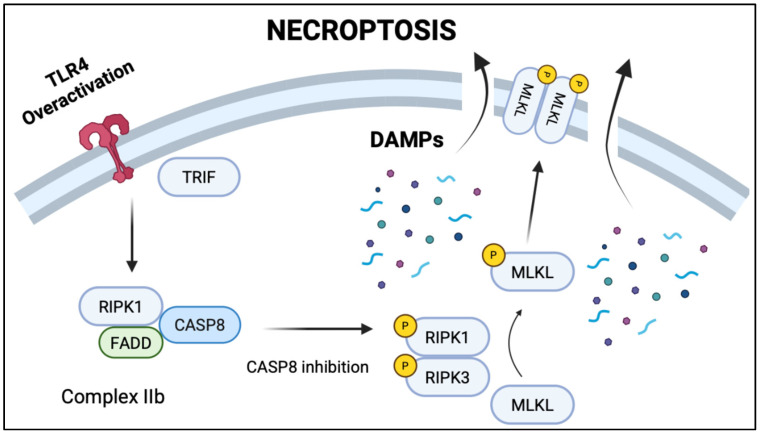
**TLR4 overactivation leading to necroptosis.** TLR4 activation triggers TRIF-mediated assembly of RIPK1, CASP8, and FADD into necroptosis complex IIb. Caspase 8 inhibition leads to the formation of RIPK1-RIPK3 complex, which recruits and phosphorylates mixed lineage kinase domain-like pseudokinase (MLKL). Phosphorylated MLKL translocates to the cellular membrane and triggers rupture and the release of damage-associated molecular patterns (DAMPs).

**Table 1 biomedicines-13-02288-t001:** Summary of the burden of NEC: incidence, disparities, and economic impact.

Parameter	Summary Data	Key Trends	References
Overall incidence	2–7% of very low birth weight (VLBW; <1500 g) infants	Higher rates in extremely preterm (<28 weeks)	[1]
Gestational age	<28 weeks: 7–10% incidence; 28–32 weeks: 2–5%; >32 weeks: <1%	Strong inverse correlation between gestational age and NEC risk	[1,45,46,47,48]
Birth weight	<1000 g: 8–12%; 1000–1500 g: 3–6%; >1500 g: <1%	Lower birth weight associated with greater severity and mortality	[45]
Mortality	15–30% overall; higher in surgical NEC (>50% in ELBW infants)	Mortality has remained high despite advances in NICU care	[1,35,49]
Racial/ethnic disparities	Higher incidence in non-Hispanic Black infants vs. White infants; disparities persist after adjusting for GA/BW	Suggests contributions from socioeconomic, healthcare access, and feeding practices	[2,35]
Long-term outcomes	20–50% risk of neurodevelopmental impairment in survivors; increased risk of short bowel syndrome and growth failure	NEC is a leading cause of post-NICU morbidity	[5,6,7,8]
Economic burden	Median additional cost per NEC case: $70,000–$180,000 USD; national annual cost >$500 million USD	Surgical NEC associated with 3–4 × higher costs than medical NEC	[1]

**Table 2 biomedicines-13-02288-t002:** Experimental NEC models: features, advantages, and limitations.

Model Type	Key Features	Advantages	Limitations	References
Small Animal Rodent Models (Mouse, Rat)	Formula feeding combined with hypoxia or cold stress, with or without LPS or NEC-causing bacteria.	Low cost, short gestation period, large litter sizes, and availability of genetic knockouts, especially mouse mutants (KO, transgenic, e.g., TLR4).	Limited resemblance to human pathophysiology, Less tissue availability for analysis	[15,18,22,87,88]
Preterm Piglet	Delivered by C-section at 90–95% gestation; fed formula with optional bacterial inoculation.	Closest to human preterm gut anatomy, physiology, and immune development; allows clinically relevant feeding studies.	High costs, specialized facilities, ethical and logistical constraints	[15,77,87,89,90]
Quails	Germ-free neonatal quails fed orally with Clostridium butyricum.	Offers a spontaneous way to study NEC, mimics key human like NEC features (cecal wall thickening, pneumatosis, hemorrhage, and mucosal necrosis).	Uncommon, species-specific immune features and anatomical differences	[91]
Rabbit	Preterm cesarean, exposure to enteral feeding, Enterobacter cloacae colonization, pharmacological agents (ranitidine, indomethacin)	Histopathological changes mirror human NEC, ranging from villous tip sloughing to transmural necrosis	Moderate survival rates, limited genetic manipulations, and more expansive compared to rodents.	[92]
Non-human Primates—rhesus macaques	Chronic intra-amniotic infection with Ureaplasma parvum induces preterm birth, systemic fetal inflammation, and NEC like pathology.	Offer unparalleled physiological relevance for studying human NEC, 95–98.5% genetic similarity to humans	Rare or nearly nonexistent, expensive, and regulatory barriers	[93,94]
Non-human Primates Baboon	Premature 125 days of gestation, 67% of the term, equivalent to 26–27 weeks in humans), managed with neonatal intensive care protocols, develop spontaneous NEC.	Offers a highly translational non-human primate system with clinical, radiologic, and histopathologic features that closely resemble those of human disease.	High cost and logistical challenges, expensive, and regulatory barriers	[95]
IEC6, Caco-2, HT-29, HTB-38	Epithelial cell lines, In vitro models treated with stimulants such as LPS, NEC bacteria, H_2_O_2_, Pro-inflammatory cytokines	Allows highly controlled treatments to study cell-specific effects.	Treatments do mimic in vivo models, low translational value	[18,96,97,98,99,100,101]
Ex vivo intestine	Isolated intestinal tissue exposed to LPS, cytokines, or formula	Allows controlled mechanistic studies; preserves tissue architecture	Short viability; no systemic immune or vascular contribution	[18,97]
Intestinal Organoids	3D cultures derived from human or animal stem cells; exposed to inflammatory stimuli.	Human-derived tissues, Scalable for multiplexing, and suitable for molecular manipulation.	Lack immune, vascular, and nervous components; immature phenotype	[84]
NEC-on-a-Chip	Microfluidic devices mimicking gut epithelium, microbiota, immune interactions	High control over microenvironment; real-time imaging; potential for personalized testing	Still experimental; may not fully replicate in vivo complexity	[86,102,103]

**Table 3 biomedicines-13-02288-t003:** Translational therapeutic strategies for NEC.

Intervention/Molecule	Preventative/Therapeutic/Mechanism of Action	Translational Stage	References
Human Breast milk/Donor milk	Mother’s own milk or donor milk provides the highest protection, naturally rich in immunomodulatory and anti-inflammatory agents (e.g., lactoferrin, sIgA, cytokines, EGF, TGF, HMOs).	Fully translational and widely accepted	[2,25,26,27,28,29,76,77]
HMOs fortification	Prebiotic effect, block pathogen adhesion, modulate immunity, inhibit TLR4 signaling	Clinical, HMOs fortified infant formulae	[25,26,27,28,29,77,78]
Probiotics (Bifidobacterium, Lactobacillus)	Modulate gut microbiota, enhance barrier function, reduce inflammation	Clinical trials (multiple RCTs)	[22,23,177]
Bovine Colostrum	Immune-modulatory proteins, growth factors, enhance barrier repair	Early-phase clinical	[79,81]
TLR4 Antagonists (e.g., Compound-34)	Block TLR4 signaling to reduce inflammation and apoptosis	Pre-clinical	[19,96,112]
Stem Cell Therapy (amniotic fluid–derived MSCs)	Promote mucosal repair, immunomodulation	Pre-clinical	[179,180,181,182]
Lactoferrin	Antimicrobial, anti-inflammatory, iron-binding	Clinical trials	[83]
Antioxidants (N-acetylcysteine, melatonin)	Reduce oxidative stress, protect mitochondria	Pre-clinical	[7,27]

## Data Availability

No data were used for the research described in the article.

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
