# Peer review of "Necrotizing Enterocolitis: A Comprehensive Review on Toll-like Receptor 4-Mediated Pathophysiology, Clinical, and Therapeutic Insights"

_biomedicines, 2025, doi:10.3390/biomedicines13092288_

Round 1

Reviewer 1 Report

Comments and Suggestions for Authors

This manuscript presents a comprehensive and well-structured review of necrotizing enterocolitis (NEC), with a central focus on Toll-like receptor 4 (TLR4)-mediated mechanisms. The authors effectively synthesize a wide range of multidisciplinary perspectives—spanning neonatology, immunology, microbiology, nutrition, and engineering—into a cohesive narrative that underscores the complexity of NEC and the value of cross-disciplinary collaboration.

The review’s strengths lie in its breadth and depth, offering detailed coverage of clinical epidemiology, racial and socioeconomic disparities, epithelial barrier dysfunction, immune dysregulation, microbial dysbiosis, and experimental models. Particularly commendable is the thorough discussion of immune cell subsets and TLR4-driven pathogenesis, which is well-supported by current literature. The inclusion of emerging concepts such as the gut-brain and gut-lung axes, epigenetic regulation, and organ-on-chip platforms further enhances its timeliness and translational appeal.

Several minor revisions are recommended to improve clarity and impact.

  • First, some sections—particularly those on experimental models and immune cell subsets—could be streamlined to reduce redundancy and enhance readability. The sections on feeding practices and epidemiology may also benefit from more concise summarization.
  • Second, while the focus on TLR4 is appropriate, expanding discussion on other innate immune pathways (e.g., NLRs, inflammasomes) would provide a more holistic view of NEC pathogenesis.
  • Finally, well-designed figures and summary tables are anticipated to aid comprehension and engagement.

In summary, this is an excellent and highly informative review that makes a significant contribution to the NEC literature. I recommend acceptance pending minor revisions as outlined above

Reviewer 2 Report

Comments and Suggestions for Authors

The review article, “Necrotizing Enterocolitis: A Comprehensive Review on Toll-like Receptor 4-Mediated Pathophysiology, Clinical, and Therapeutic Insights,” addresses a critical neonatal pathology that continues to have high mortality rates among premature infants, largely due to its complex pathophysiology. While the authors aimed to cover nearly all aspects of the disease, their approach faces a key challenge: balancing comprehensiveness with conciseness to avoid lengthy, less informative sections that might deter readers. Additionally, a crucial consideration for any newly published review is the inclusion of up-to-date information and original insights in the authors’ analysis. In my view, the manuscript has weaknesses in both these areas and requires to be revised.

Further I will try to explain my opinion.

- The manuscript lacks a clearly defined target audience, and its novelty within the current scientific landscape is unclear. Without a focused approach, the manuscript reads more like a compilation of information from existing, more specialized reviews. The role of TLR4-mediated stimulation in NEC has already been extensively reviewed (e.g., 10.1016/j.jcmgh.2018.04.001; 10.1016/j.clp.2018.09.007). Similarly, animal models of NEC have been covered in detail (10.1515/iss-2017-0050; doi.org/10.3389/fimmu.2024.1434281). Additionally, numerous other reviews discuss NEC pathology (10.1007/s00383-023-05619-3; 10.1038/s41575-022-00594-x; 10.3389/fped.2022.1107404).

I suggest using the main idea presented in the Introduction as the scope of the manuscript—a multidisciplinary approach to understanding NEC disease. By incorporating more up-to-date evidence and fresh conclusions, I believe this review will demonstrate originality while allowing the authors to address pathophysiological, clinical, and therapeutic aspects comprehensively. Given the authors' profound expertise in the clinical aspects of NEC, this could serve as the central focus of the review.

-The current structure appears suboptimal. It is very difficult to follow the logic due to the numerous subheadings in the text. Splitting the 'Introduction' section into number of small paragraphs to partially address the causes, diagnostic methods, and therapeutic approaches of the disease is unconventional (these sections are better to be used in main part). A more effective approach would be to follow the classic structure: first providing essential background information on the disease, then outlining the current state of knowledge and key challenges, and finally leading into the main focus of the article.

- Beyond the Introduction, other sections appear excessive and may warrant reconsideration. For instance, Section 6 ('Clinical Treatment Strategies for NEC') contains only two subheadings, while Section 4 ('Prevention, Treatment, and Translational Advances') is far more extensive, with six subsections. To improve structural balance, these sections could potentially be consolidated—for example, by integrating current conventional approaches with emerging (but not yet implemented) strategies under a unified chapter.

- It is unclear why the 'Experimental Models' subsection is grouped under 'Pathogenesis and Mechanistic Insights' alongside other subheadings like 'Immature Intestinal Barrier.' Given the manuscript’s title—'Necrotizing Enterocolitis: A Comprehensive Review on Toll-like Receptor 4-Mediated Pathophysiology, Clinical, and Therapeutic Insights'—the extensive discussion of experimental NEC models even seems excessive.

Specific considerations:

  1. “Aims of the Review” in Abstract seems to be excessive.
  2. Since birth weight and time of childbirth are the main risk factors for NEC, in the 'Future Directions and Conclusions' section, I suggest addressing the issue how to maintain the full-term pregnancy, as this may help reduce NEC incidence
  3. The authors note that racial and ethnic disparities influence NEC incidence. To offer readers a more comprehensive perspective—mainly to avoid implying that genetics alone contributes to NEC development—I suggest including a more detailed explanation where racial and ethnic disparities also include differences in lifestyle, unequal access to resources, opportunities, and medical treatment.
  4. Paragraph “2.3. NEC and Birth Weight” is more focused on geographical and economic issues rather than weight correlations. If analysis of reasons that effect NEC incidence is in the focus of the article it deserves also to be better structured.
  5. The authors state in “3. Pathogenesis and Mechanistic Insights” that the initial trigger for NEC is believed to involve pathogenic bacterial invasion into the immature intestinal barrier, leading to a dysregulated pro-inflammatory immune response (lines 356–358). While this reasoning has merit, it appears to be an oversimplification. Research has demonstrated that NEC can develop even in the absence of pathogenic bacteria—for example, through the cold/hypoxia method (Nat Commun 2020, doi:10.1038/s41467-020-19400-w). A second proposed mechanism involves the highly permeable immature intestinal barrier, which, due to insufficient regulatory T-cell (Treg) suppression, becomes inflamed simply in response to pathogen-associated molecular patterns (PAMPs) from commensal bacteria in the lumen. The authors mention this mechanism further, but upon reading, “pathogenic bacteria” mechanism appear to be the only one—or at least the main.

Reviewer 3 Report

Comments and Suggestions for Authors

This article is a comprehensive review of Necrotizing Enterocolitis (NEC). NEC is one of the most urgent gastrointestinal emergencies in neonates, and therefore, establishment of effective preventive and therapeutic strategies is crucial. The article provides a foundation of knowledge on NEC. This reviewer is concerned with the following points to improve the quality of the content.

  1. While the authors describe that TLR4 is a central driver of NEC, the connection between TLR4 and the disease is not sufficiently explained and requires more detail.
  2. Section 2.4. Feeding Intolerance and many other sections: The descriptions are too long and should be divided into several paragraphs.
  3. Section 3. Pathogenesis and Mechanistic Insights: Please include figures to visualize the pathogenesis of NEC.
  4. Please include figures in other sections as well, to improve the clarity of the presentation
  5. Section 3.1. Experimental Models of NEC: It is difficult to find how the described animal models mirror the pathological features of NEC. Please explain the models in more detail.
  6. Sections 3.5. Immune Landscape in NEC and 3.7. Oxidative Stress and Cell Death: please indicate whether the referenced studies were based on animal models, human cells or clinical samples. It also applies for other sections.
  7. Section 4. Prevention, Treatment, and Translational Advances: Please create a table and indicate which translational state (pre-clinical or clinical trials) the molecules described in this section are in.
  8. Please add a Table listing medicines currently used for treatment of NEC.

Reviewer 4 Report

Comments and Suggestions for Authors
  1. The "Burden of NEC" section (1.1) is data-dense. Consider condensing epidemiological statistics into a table and focusing on key trends (e.g., racial disparities, cost burden).
    1. Subsections (3.1–3.8) are detailed but occasionally repetitive (e.g., TLR4’s role is reiterated in "Immature Intestinal Barrier," "TLR4 Signaling," and "Immune Landscape").
    2. Define abbreviations at first use (e.g., PPD, HMOs, NETs).
    3. Clarify distinctions between NEC and spontaneous intestinal perforation (SIP) earlier in the text (currently in Section 2.5).
    4. Some pathways (e.g., necroptosis in Section 3.7.3) are described technically. Add simplified schematics or analogies for broader readability.
    5. Expand on clinical trial challenges (e.g., safety of TLR4 inhibitors in neonates, probiotic strain selection).
    6. Add a sentence on the review’s translational impact (e.g., "This review bridges mechanistic insights to clinical applications, advocating for personalized NEC prevention")in asbtract portion
  1. include a flowchart of NEC pathogenesis centered on TLR4.
  2. A table comparing experimental models (advantages/limitations).
  • A summary table of therapeutic strategies (mechanism, evidence level).
  1. Strengthen the "Future Directions" section by prioritizing 3–5 actionable goals (e.g., biomarker validation, microbiome engineering).

Round 2

Reviewer 2 Report

Comments and Suggestions for Authors

I appreciate the authors’ efforts in revising the manuscript. It has now become a comprehensive scientific work, approaching the scale of a monograph. While its breadth is commendable, the level of detail remains insufficient for a highly-cited publication.

1. The Introduction section remains suboptimal. While it has been stripped of its numerical ordering, the core content is unchanged and the red colored text is still more suitable for a dedicated chapter arguing the necessity of a multidisciplinary approach to diagnosing NEC, studying its pathogenesis, and developing therapies. Although the introduction includes epidemiological data (e.g., incidence and prevalence), it lacks other critical elements expected by readers. Crucially, it fails to articulate the novel contribution this review makes to the existing body of knowledge. Basically, it lacks the justification for why this review needs to be published. If the authors posit the multidisciplinary framework as their innovative perspective, they must not only rationalize this choice but also explicitly describe the potential advantages and outcomes of applying such an approach. A more effective introduction should provide Background & Current Understanding, Research Gap or Problem, Purpose & Aim of the Review, Scope of the Review and Roadmap.

  1. Overall, the structure of the review article has become clearer. To facilitate navigation, I would recommend keeping at least the main chapter numbering: 1. Clinical Aspects of NEC, 2. Clinical Treatment Strategies, 3. Pathogenesis and Mechanistic Insights, etc.

However, the text structure still requires revision. The chapter "Pathogenesis and Mechanistic Insights" begins with the subchapter "Experimental Models to Study NEC Pathogenesis" before addressing more specific sections directly related to pathogenesis, such as "Pathogenesis and Mechanistic Insights". To improve readability, I would suggest either making "Experimental Models to Study NEC Pathogenesis" a following separate chapter or beginning the "Pathogenesis" chapter with the more specialized subchapters, such as "Compromised Gut Epithelium..." and others. The section on experimental models could then follow, noting that the aforementioned pathogenic mechanisms can be reproduced in vivo and in vitro.

  1. The authors have done considerable work in presenting the figures and tables to clarify the complex relationships between different factors in NEC. However, in my view, these figures require amendment.

- Specifically, the graphical abstract is overly complex due to the numerous intercrossing arrows. I would recommend a revised structure: place the human factors and NEC models in the upper section as the starting point. The middle section should feature TLR4 overstimulation, from which descending arrows would illustrate the subsequent pathological processes (e.g., barrier dysfunction, cell death, ENS dysfunction). The therapeutic strategies could then be presented in a separate box, as currently shown.

This revised schematic would provide a more logical flow for understanding the causes and consequences of NEC."

- Fig. 1 contains the odd slash mark in the phrase 'premature intestine'.

- Figure 3 appears to be a specific case of Figure 2, which seems redundant. The text does not make it clear why the authors have chosen to detail necroptosis specifically, and not other forms of cell death also implicated in NEC (such as apoptosis, pyroptosis, autophagy, and ferroptosis) [10.3390/ijms26094036, 10.3389/fped.2023.1199878]. If the authors wish to provide a more detailed mechanism of cell death in NEC, it would be more logical to expand Figure 3 to include these pathways

- In Table 2, the 'Non-human Primates and Baboons' lines, the authors probably meant to use 'expensive' instead of 'expansive.' If 'expansive' was indeed the intended word, its meaning should be explained more clearly

- In table 2 it is also not clear why baboons are mentioned separately from NHPs.

- Reference 94 does not cover NHP-based models of NEC

4.The 'Future Directions and Conclusions' section lacks references to prospective diagnostic and therapeutic approaches, except in the final chapter. It is unclear whether the directions outlined are the authors' original proposals or if they have been previously published. To strengthen this chapter, it should revisit the aims stated in the Introduction and elaborate on the potential advantages and outcomes of employing a multidisciplinary strategy.

5. In the abstract the word 'advocating' is repeated in the next two sentences. Please rephrase

Reviewer 3 Report

Comments and Suggestions for Authors

The authors revised manuscript well. However, this reviewer still concerns figures. Figure 1 and figure 3 look similar. The TLR4-mediated signal transduction pathway leading to necroptosis should be depicted in more detail in Figure 3. 

Author Response

Thank you very much for reviewing our revised manuscript. We appreciate that the reviewer agreed with our revisions. Based on the reviewer's concerns and suggestions, we have revised the necroptosis figure (Figure 3).

Round 3

Reviewer 2 Report

Comments and Suggestions for Authors

The authors have done valuable work. I believe the manuscript is suitable for publication in its present form

Reviewer 3 Report

Comments and Suggestions for Authors

The authors replied to comment by this reviewer.  The figure 3 of this manuscript is better now.